# A SIMPLE CONTRACTION CRITERION FOR THE SINKHORN MIRROR FLOW

## ABSTRACT

We give a concise condition for contraction of the continuous-time mirror dynamics which was recently shown to be the vanishing-step-size limit of the Sinkhorn algorithm. This condition is essentially coercivity of a conditional expectation operator.

## 1 INTRODUCTION

Recently, entropy-regularized optimal transport and the closely related Schrödinger bridge problem Léonard (2013) have found theoretically attractive and experimentally competitive applications in generative modeling De Bortoli et al. (2021). Its increasing usage and analysis is due in large part to the use of the Sinkhorn algorithm for entropy-regularized OT Cuturi (2013). Recent work Karimi et al. (2024) has shown that the limit of vanishing step-sizes of Sinkhorn is a mirror-gradient flow, and provide asymptotic convergence rates of the objective function. Here we leverage methods from contraction analysis Wensing & Slotine (2020) to give criteria for contraction (exponential pairwise convergence of two particular solutions) of this flow as a coercivity condition on a certain conditional expectation operator.

## 2 BACKGROUND

### 2.1 SCHRÖDINGER BRIDGE PROBLEM

Recall that the *Schrödinger bridge problem* (in hydrodynamic form) is

$$\inf_{\boldsymbol{u},\rho} \int_0^1 \mathcal{K}(\boldsymbol{u_t}, \rho_t)dt \quad \text{where } \mathcal{K}(\boldsymbol{u}, \rho) = \int_{\mathbb{R}^d} |\boldsymbol{u}(x)|^2 \rho(x)dx \tag{1}$$

$$\text{s.t. } \partial_t \rho + \nabla \cdot \rho_t \boldsymbol{u}_t = \varepsilon \Delta \rho_t, \quad \rho_0 \text{ and } \rho_1 \text{ given} \tag{2}$$

for prescribed initial and terminal densities $\rho_0, \rho_1$. (We work exclusively with hydrodynamic rather than stochastic control forms. The control formulation illuminates that problem 1 is that of attaining a bridge $\boldsymbol{X}_t$ between $\rho_0, \rho_1$ with $\text{Law}(\boldsymbol{X}_t) = \rho_t$ which is closest in the sense of $\mathcal{K}$ to a reference diffusion, here the Brownian motion.) By introducing a change of variables $\boldsymbol{v} = \boldsymbol{u} - \varepsilon \nabla \log \rho$, it was shown in Chen et al. (2016) that system 1, 2 is equivalent to the alternative hydrodynamic form

$$\inf_{\boldsymbol{v},\rho} \int_0^1 \left[ \mathcal{K}(\boldsymbol{v}_t, \rho_t) + \frac{\varepsilon}{4} \mathcal{F}(\rho_t) \right] dt, \quad \text{where } \mathcal{F}(\rho) = \int_{\mathbb{R}^d} |\nabla \log \rho(x)|^2 \rho(x)dx \tag{3}$$

$$\text{s.t. } \partial_t \rho + \nabla \cdot \rho \boldsymbol{v} = 0, \quad \rho_0 \text{ and } \rho_1 \text{ given}, \tag{4}$$

showing that the Schrödinger bridge problem is an $\varepsilon$-regularization of Benamou & Brenier (2000)'s formulation of optimal transport by the Fisher information functional $\mathcal{F}$. Introducing $\lambda$ as a Lagrange multiplier enforcing equation 4, it can be shown that $\boldsymbol{v}$ is of gradient form $\boldsymbol{v} = \nabla \lambda$ where $\lambda$ solves the Hamilton-Jacobi equation

$$\partial_t \lambda + \frac{1}{2} ||\nabla \lambda||^2 = 0. \tag{5}$$

Let $\mu \in \mathcal{P}(X), \nu \in \mathcal{P}(Y)$ be target marginals for a plan in $\pi \in \Pi(\mu, \nu)$, whose actual marginals we will denote by $\pi^X, \pi^Y$. Recall that the *entropy-regularized* optimal transport problem is

$$\text{OT}_\varepsilon = \inf_{\pi \in \Pi(\mu,\nu)} \left[ \mathbb{E}_\pi[c] + \varepsilon H(\pi || \mu \otimes \nu) \right]. \tag{6}$$

Here, and throughout, we will denote the sets with one-sided marginal constraints as

$$\Pi(\mu, \cdot) = \{\pi \in \mathcal{P}(X \times Y) \mid \pi^X = \mu\}, \ \Pi(\cdot, \nu) = \{\pi \in \mathcal{P}(X \times Y) \mid \pi^Y = \nu\} \tag{7}$$

for convenience. A remarkable fact is that Schrödinger bridge problem 1, 2 is equivalent to 6. An iterative scheme for solving 6 is the *Sinkhorn algorithm*

$$d\pi_0 \propto \exp(-c/\varepsilon)d(\mu \otimes \nu) \tag{8}$$

$$\pi_{n+1} = \arg\min_{\pi \in \Pi(\mu, \cdot)} H(\pi || \pi_n) \tag{9}$$

$$\pi_{n+2} = \arg\min_{\pi \in \Pi(\cdot, \nu)} H(\pi || \pi_{n+1}) \tag{10}$$

which iterates a minimizing-entropy scheme with fixed endpoints $\mu, \nu$ at the odd and even steps respectively.

## 2.2 SINKHORN FLOW AS MIRROR DESCENT

It has recently been shown in Karimi et al. (2024) that the Sinkhorn algorithm 8 admits a continuous-time formulation as a mirror descent, where

$$\frac{\partial h_t}{\partial t} = -\frac{\delta F}{\delta \pi}(\pi_t) = -\log\frac{d\pi_t^Y}{d\nu}, \quad F(\pi) = H(\pi^Y || \nu), \ h_t \in L^1(X \times Y) \tag{11}$$

is the flow in the dual space $L^1(X \times Y)$, and

$$\pi_t = \frac{\delta\varphi^*}{\delta h}(h_t) = \hat{\pi}, \quad \varphi^*(h) = \langle\hat{\pi}, h\rangle - H(\hat{\pi} || \pi_0), \ \hat{\pi} = \frac{\pi_0(x, y)e^{h(x,y)}\mu(x)}{\int_{\mathbb{R}^d}\pi_0(x, y')e^{h(x,y')}dy'} \tag{12}$$

is the flow in the primal space $\mathcal{P}(X \times Y)$, noting that $\hat{\pi}^X = \mu$ by construction. Here, $F$ is the *objective* and $\varphi$ the *mirror map*, and $\varphi^*$ is the Legendre transform (Fenchel conjugate) of the mirror map $\varphi$

$$\varphi^*(h) = \sup_{\pi \in \Pi(\mu, \cdot)} \langle\pi, h\rangle - \varphi(\pi), \quad \text{where } \varphi(\pi) = H(\pi || \pi_0) + \iota_{\Pi(\mu, \cdot)}(\pi), \tag{13}$$

which is shown to have the form 12 by Lemma 3, Karimi et al. (2024).

## 2.3 DEFINITIONS

**Definition 2.1** (Conditional expectation operator). Define

$$(P_\pi f)(y) := \mathbb{E}_\pi[f|Y = y] = \frac{1}{\pi^Y(y)}\int_X f(x, y)\pi(x, y)dx, \tag{14}$$

which is an orthogonal projection on $L^2(\pi)$, since, for all $f, g \in L^2(\pi)$,

1. $P_\pi$ is a projection by the tower property

$$P_\pi P_\pi f = \mathbb{E}_\pi[\mathbb{E}_\pi[f|Y]|Y] = E_\pi[f|Y] = P_\pi f, \tag{15}$$

2. $P_\pi$ is self-adjoint by

$$\langle P_\pi f, g\rangle_{L^2(\pi)} = \iint_{X \times Y} \frac{g(x, y)\pi(x, y)}{\pi^Y(y)}dx\int_X f(x', y)\pi(x', y)dx'dy = \langle f, P_\pi g\rangle_{L^2(\pi)}. \tag{16}$$

3. $P_\pi$ is the orthogonal projection onto the closed subspace

$$L_Y^2(\pi) := \{g \in L^2(\pi) \mid g(x, y) = g(y) \text{ a.e.}\} \tag{17}$$

whose orthogonal complement is

$$\ker P_\pi = \{f \in L^2(\pi) \mid E_\pi[f|Y] = 0 \text{ a.e.}\} \tag{18}$$

## 3 MAIN RESULTS

**Theorem 3.1** (Contraction of Sinkhorn flow). *The Sinkhorn mirror flow 11, 12 is contracting (or expanding) with rate $\lambda \in \mathbb{R}$ in the metric*

$$\langle \frac{\eta}{\pi_t}, \frac{\xi}{\pi_t} \rangle_{L^2(X \times Y)} \tag{19}$$

*for all states $\pi_t$ at which the conditional expectation operator $P_{\pi_t}$ defined in 14 satisfies the coercivity property*

$$\langle \xi^Y, P_{\pi_t} \xi \rangle_{L^2(Y)} \geq 2\lambda \langle \xi, \xi \rangle_{L^2(X \times Y)} \tag{20}$$

*Proof.* Let $\xi \in L^1(X \times Y)$. The Gateaux derivative (first variation) of $F$ at $\pi$ in the direction $\xi$ is

$$\delta F(\pi)[\xi] = \iint_{X \times Y} \left( \log \frac{\pi^Y(y)}{\nu(y)} + 1 \right) \xi(x,y) dx dy = \int_Y \xi^Y(y) \left( \log \frac{\pi^Y(y)}{\nu(y)} + 1 \right) dy, \tag{21}$$

where $\xi^Y(y) = \int_X \xi(x,y) dx$ is the "marginal." Similarly, the second variation is, given also $\eta \in L^1(X \times Y)$,

$$\delta^2 F(\pi)[\xi, \eta] = \int_Y \frac{\xi^Y(y)\eta^Y(y)}{\pi^Y(y)} dy \tag{22}$$

and in particular we have the positive-semidefiniteness

$$\delta^2 F(\pi)[\xi, \xi] = \int_Y \frac{(\xi^Y(y))^2}{\pi^Y(y)} dy \geq 0, \text{ with equality iff } \xi^Y = 0 \text{ a.e.} \tag{23}$$

Next, we consider the mirror map $\varphi$. Let us define first the tangent space

$$T_\pi \Pi(\mu, \cdot) = \{a \in L^1(X \times Y) \mid a^X(x) = 0 \text{ for } \mu - \text{a.e. } x\} = \{a \in L^1 \mid \iint_{X \times Y} a = 0\}. \tag{24}$$

Then

$$\delta \varphi(\pi)[a] = \iint_{X \times Y} \left( \log \frac{d\pi}{d\pi_0} + 1 \right) a \, dA, \quad a \in T_\pi \Pi(\mu, \cdot) \tag{25}$$

and

$$\delta^2 \varphi(\pi)[a, b] = \iint_{X \times Y} \frac{ab}{\pi} dA, \quad a, b \in T_\pi \Pi(\mu, \cdot) \tag{26}$$

which is strictly positive definite since $\pi > 0$ a.e. Now, let $\delta\pi_t \in T_\pi \Pi(\mu, \cdot)$ be a perturbation. Since

$$h_t = \delta\varphi(\pi_t) \quad \text{and} \quad \frac{d}{dt} h_t = -\delta F(\pi_t) \tag{27}$$

then $\delta\pi_t$ induces a corresponding $\delta h_t \in L^1$ by 26 as

$$\delta h_t = \delta^2 \varphi(\pi_t)[\delta\pi_t, \cdot] = \frac{\delta\pi_t}{\pi_t} \quad \text{and} \quad \frac{d}{dt} \delta h_t = -\delta^2 F(\pi_t)[\delta\pi_t, \cdot] = -\frac{\delta\pi_t^Y}{\pi_t^Y} \tag{28}$$

which is well-defined since $\pi_t^Y > 0$ a.e. Note then that the Hessian operators of $\varphi, F$ are expressible as

$$H_t^\varphi \delta\pi_t = \frac{\delta\pi_t}{\pi_t}, \quad H_t^F \delta\pi_t = P_{\pi_t} H_t^\varphi \delta\pi_t. \tag{29}$$

It follows from 28 and the definition 14 of $P_\pi$ that

$$P_{\pi_t} \delta h_t = P_{\pi_t} \frac{\delta\pi_t}{\pi_t} = \frac{1}{\pi_t^Y} \int_X \delta\pi_t(x,y) dx = \frac{\delta\pi_t^Y}{\pi_t^Y} = -\frac{d}{dt} \delta h_t. \tag{30}$$

The metric in the hypothesis is

$$M_t = (H_t^\varphi)^2 \tag{31}$$

which is valid since $H^\varphi = (H^\varphi)^*$. The norm of the perturbation in this metric evolves as

$$\frac{d}{dt}\frac{1}{2}\|\delta\pi_t\|_{M_t}^2 = \frac{d}{dt}\frac{1}{2}\|\delta h_t\|_{L^2}^2 = \langle \delta h_t, \frac{d}{dt}\delta h_t\rangle = -\delta^2 F(\pi_t)[\delta\pi_t, \delta h_t] \tag{32}$$

$$= -\langle \frac{\delta\pi_t^Y}{\pi_t^Y}, \left(\frac{\delta\pi_t}{\pi_t}\right)^Y\rangle_{L^2(Y)} = -\langle P_{\pi_t}\frac{\delta\pi_t}{\pi_t}, \left(\frac{\delta\pi_t}{\pi_t}\right)^Y\rangle \tag{33}$$

$$\leq -2\lambda\|\delta\pi_t\|_{M_2}^2 \tag{34}$$

by hypothesis, using the change of variable $\xi = \delta\pi_t/\pi_t$. □

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
