# OpenReview forum: "A simple contraction criterion for the Sinkhorn mirror flow"
_ICLR.cc/2026/Conference — ICLR 2026 Conference Withdrawn Submission_

### Official Review · Reviewer_KSwB · 2025-10-30

**Soundness:** 3
**Presentation:** 3
**Contribution:** 2
**Rating:** 2
**Confidence:** 4

**Summary:**

The primary contribution of the paper is to show that coercivity of the conditional expectation operator is a condition for contraction of continuous time mirror dynamics of the sinkhorn flow - a method to compute the entropy-regularized optimal transport plan.

**Strengths:**

The paper is original and clear and seems to address a theoretical gap in analysis of the celebrated Sinkhorn alg. for entropy-reg OT.

**Weaknesses:**

The only main contribution of the paper is Theorem 3.1, which seems to be novel, but I was not able to appreciate if there are new proof techniques that were introduced, whether it solves something that was previously hard to understand, and what are the consequences of the Theorem.

**Questions:**

It will be helpful if the authors can clarify the utility of the Theorem - what are the consequences of this result and why this result deserves attention by the ML community - applied and theoretical.

---

### Official Review · Reviewer_Pqzx · 2025-11-01

**Soundness:** 1
**Presentation:** 1
**Contribution:** 1
**Rating:** 0
**Confidence:** 5

**Summary:**

This seems a unfinished draft/

**Strengths:**

N/A

**Weaknesses:**

N/A

**Questions:**

N/A

---

### Official Review · Reviewer_x8ti · 2025-11-03

**Soundness:** 2
**Presentation:** 1
**Contribution:** 1
**Rating:** 2
**Confidence:** 4

**Summary:**

This paper provides a simple contraction condition for the continuous-time limit of the Sinkhorn algorithm, namely the coercivity of a conditional expectation operator.

**Strengths:**

.

**Weaknesses:**

The paper could better motivate the importance of the contraction condition and provide a clearer background discussion. For instance, Section 2.2 is relatively too short compared to Karimi et al. (2024). I recommend expanding the paper to make it more accessible and informative to readers. In its current form, the paper appears too preliminary and is not ready for publication.

**Questions:**

.

---

### Official Review · Reviewer_NaUm · 2025-11-03

**Soundness:** 3
**Presentation:** 2
**Contribution:** 2
**Rating:** 2
**Confidence:** 3

**Summary:**

This paper establishes a contraction criterion for the continuous-time mirror dynamics that emerges as the vanishing step-size limit of the Sinkhorn algorithm (which was shown in prior work). The authors provide a sufficient condition for exponential contraction (or expansion) of the Sinkhorn mirror flow. This condition is expressed as a coercivity property of the conditional expectation operator $P\_\pi$, where $\pi$ is the variable denoting the measure which we're optimizing. The paper is quite brief (4 pages including the references), its contribution is a single theorem on the continous-time limit of Sinkhorn iteration.

**Strengths:**

1. The paper is on point and it introduces the setting and the necessary assumptions quite well.
2. OT at least to be quite a hot topic for ML applications such as computer vision, though it is less talked about these days.

**Weaknesses:**

1. This paper doesn't seem like a good for ICLR. The authors do not discuss a single thing related to machine learning.
2. To the best of my knowledge, the continuous time limit of Sinkhorn iteration, while interesting theoretically, is not directly relevant to any applications.
3. I do not actually see what insight we gain from the new coercivity property.

**Questions:**

1. Could the authors please comment on the significance of their results?
2. Why did the authors decide to submit to ICLR instead of a journal/conference on mathematics/optimal transport?

---

### Note · Authors · 2025-11-12

**Comment:**

Withdrawing in favor of an updated, more complete manuscript which we have prepared. We thank the reviewers for their feedback and are incorporating comments, particularly with regard to improved background / motivation and discussion of the practical relevance of the results, which we show but could not be included in this version.

**Withdrawal Confirmation:**

I have read and agree with the venue's withdrawal policy on behalf of myself and my co-authors.